# *TBX3* and *EFNA4* Variant in a Family with Ulnar-Mammary Syndrome and Sagittal Craniosynostosis

**DOI:** 10.3390/genes13091649

**Published:** 2022-09-14

**Authors:** Moon Ley Tung, Bharatendu Chandra, Jaclyn Kotlarek, Marcelo Melo, Elizabeth Phillippi, Cristina M. Justice, Anthony Musolf, Simeon A. Boyadijev, Paul A. Romitti, Benjamin Darbro, Hatem El-Shanti

**Affiliations:** 1Stead Family Department of Pediatrics, Division of Medical Genetics and Genomics, University of Iowa Carver College of Medicine, Iowa City, IA 52242, USA; 2Department of Medicine, National University of Singapore, Singapore 117597, Singapore; 3Genometrics Section, Computational and Statistical Genomics Branch, National Human Genome Research Institute, National Institutes of Health (NIH), Baltimore, MD 20892, USA; 4Statistical Genetics Section, Computational and Statistical Genomics Branch, National Human Genome Research Institute, National Institutes of Health (NIH), Baltimore, MD 20892, USA; 5Department of Pediatrics, University of California Davis, Sacramento, CA 95616, USA; 6Department of Epidemiology, College of Public Health, University of Iowa, Iowa City, IA 52242, USA

**Keywords:** Ulnar Mammary Syndrome, *TBX3*, *EFNA4*, sagittal craniosynostosis

## Abstract

Ulnar-mammary syndrome (UMS) is a rare, autosomal dominant disorder characterized by anomalies affecting the limbs, apocrine glands, dentition, and genital development. This syndrome is caused by haploinsufficiency in the T-Box3 gene (*TBX3*), with considerable variability in the clinical phenotype being observed even within families. We describe a one-year-old female with unilateral, postaxial polydactyly, and bilateral fifth fingernail duplication. Next-generation sequencing revealed a novel, likely pathogenic, variant predicted to affect the canonical splice site in intron 3 of the *TBX3* gene (c.804 + 1G > A, IVS3 + 1G > A). This variant was inherited from the proband’s father who was also diagnosed with UMS with the additional clinical finding of congenital, sagittal craniosynostosis. Subsequent whole genome analysis in the proband’s father detected a variant in the *EFNA4* gene (c.178C > T, p.His60Tyr), which has only been reported to be associated with sagittal craniosynostosis in one patient prior to this report but reported in other cranial suture synostosis. The findings in this family extend the genotypic spectrum of UMS, as well as the phenotypic spectrum of *EFNA4*-related craniosynostosis.

## 1. Introduction

Ulnar-mammary syndrome (UMS, OMIM #181450) is an autosomal dominant disorder with considerable phenotypic heterogeneity that ranges from dorsalization of the little finger [1] to complete absence of the forearm but classically presents as posterior upper limb deficiencies (ulnar ray). The clinical spectrum includes apocrine gland malformations and dysfunction, dental anomalies, delayed puberty, abnormal external genitalia in males, imperforate hymen in females, ventricular septal defects, arrhythmias, subglottic stenosis, pyloric stenosis, and anal atresia or stenosis [2,3,4,5,6]. Approximately 128 cases of UMS have been published in the literature [Orphanet Home Page. Available online: https://www.orpha.net/consor/cgi-bin/Disease_Search.php?Ing=EN&data_id=2808&Disease_Search_diseaseGroup=Ulnar-mammary-syndrome&Disease_Disease_Search_diseaseType=Pat&Disease(s)/group/group%20of%20diseases=Ulnar-mammary-syndrome&title=Ulnar-mammary%20syndrome&search=Disease_Search_Simple (accessed on 31 August 2022)]. UMS has been postulated to be caused by haploinsufficiency in the T-Box 3 gene (*TBX3*, OMIM * 601621) [2,3,4,6,7], although alternative mechanisms, such as nonsense-mediated mRNA decay [8] and dominant negative effects [5,9], have been proposed.

Craniosynostosis is a birth defect where the skull bones of a newborn close prematurely and is the second most common group of craniofacial anomalies [10]. The estimated prevalence of craniosynostosis in approximately 1 in 2500 births, worldwide [11]. Although more than one hundred genetic syndromes that include craniosynostosis as a main clinical feature have been described, about two-thirds of individuals with craniosynostosis present as nonsyndromic [12]. Nonsyndromic craniosynostosis is clinically heterogenous involving one or more of the four main sutures, with about one-half of individuals presenting with sagittal craniosynostosis [10]. The genetic etiology of nonsyndromic craniosynostosis is unclear although rare variants in a handful of genes, such as *FGFR2*, *TWIST1*, *FREM1*, *LRIT3*, *EFNA4*, and *RUNX2,* have been reported in a minority of these patients [13].

## 2. Detailed Case Description

A one-year-old, Northern European female was referred for genetic evaluation due to unilateral postaxial polydactyly and bilateral abnormal fifth fingernails. She was born at term to a 34-year-old nulliparous woman, by cesarean section due to failure to progress. Pregnancy was complicated by maternal gestational diabetes. Her growth parameters were within the normal limits. There was hypoplasia of the distal phalanx with duplication of the nail of bilateral fifth fingers, and postaxial polydactyly of the right hand (Figure 1C,D).

The proband’s father had bilateral contractures of the fifth fingers and bilateral duplication of the fifth fingernail without polydactyly, since birth. Recently, he had surgery to remove the duplicated nail in one hand (Figure 1A,B). He also had a history of congenital sagittal craniosynostosis that was surgically repaired, small nipples, asymmetric chest hair growth, and pectus excavatum.

A targeted next-generation sequencing (NGS) of 31 genes (custom panel) was performed on the proband in a CLIA-certified laboratory, which included genes known to be associated with UMS and craniosynostosis (Appendix A). Sequencing revealed a heterozygous, likely pathogenic, variant in *TBX3* (NM_005996.3: c.804 + 1G > A: IVS3 + 1G > A) that was inherited from her father. This variant is not observed in a large population cohort (gnomAD v.2.1.1) and is not reported in the ClinVar database. The precomputed, in silico model predicted this canonical splice site variant in intron 3 of *TBX3* to result in an in-frame deletion of a critical region. The SpliceAI algorithm predicts this variant to result in a donor or acceptor loss (Appendix A).

The proband’s dizygotic twin brothers were evaluated at 3 weeks of age and only one of them had a unilateral, stiff fifth finger on physical examination (Figure 1E). However, both of these twin brothers were also heterozygous for the same *TBX3* variant.

In view of the additional clinical findings of sagittal craniosynostosis in the proband’s father, we performed genome sequencing on him which revealed a heterozygous, missense variant in *EFNA4* (NM_005227.3: c.178C > T, p.His60Tyr) (Appendix A). This variant was predicted to be damaging on Mutation Taster and has a combined annotation dependent depletion (CADD) score of 25.8. The minor allele frequency of this variant was 0.001346 with 79 carriers and one homozygote as reported by gnomAD v3.1.2. WGS also identified the *TBX3* variant. Targeted Sanger sequencing in the proband and her twin brothers did not detect this *EFNA4* variant (Appendix A). The proband’s mother was negative for both the *TBX3* and *EFNA4* variants.

## 3. Genetics Testing

### Materials and Methods

Whole genome sequencing (WGS) was performed by the Iowa Institute of Human Genetics Genomic Division laboratory using manufacturer-recommended protocols. Briefly, one microgram of genomic DNA was sheared using the E220 focused-ultrasonicator (Covaris) to give an average fragment length of 400 bp. Indexed sequencing libraries were generated by using the KAPA HyperPrep kit for Illumina sequencing (Cat. No. KK8504, Roche/Kapa Biosystems (Roche Sequencing and Life Science, Kapa Biosystems, Wilmington, NC, USA), Roche Sequencing and Life Science, Kapa Biosystems) with three PCR cycles of PCR. The libraries were then evaluated for quantity and size using the fragment analyzer running the NGS Fragment Kit (Agilent Technologies, Santa Clara, CA, USA) and pooled based on molarity to give the desired coverage depth. The pooled libraries were sequenced on an Illumina NovaSeq 6000 S4 flow cells using 150 bp paired-end SBS sequencing chemistry (v1.5). WGS data were processed using the Illumina DRAGEN v3.9 germline DNA pipeline. Annotation and variant filtering were performed in VarSeq (Golden Helix, Bozeman, MT, USA). The combined annotation dependent depletion (CADD) score and in silico prediction models (DANN, DEOGEN2, EIGEn, FATHMM-MKL, LIST-S2, M-Cap, MVP, Mutation Assessor, MutationTaster, and SIFT) were utilized as computational evidence of the deleteriousness of the detected variants. Splice site variants were further annotated using the SpliceAI algorithm.

## 4. Discussion

We report a female child with a UMS phenotype characterized by bilateral hypoplasia of the distal phalanx of the fifth finger and duplication of the nail, unilateral postaxial polydactyly. This was associated with a heterozygous, likely pathogenic splice site variant in *TBX3* (c.804 + 1G > A; IVS3 + 1G > A). Her father and both dizygotic twin brothers harbor the same *TBX3* variant with remarkable intrafamilial variability. 

The T-Box 3 gene (*TBX3*) is part of the T-box transcription family. *TBX3* is highly conserved across a wide spectrum of species [14], and functions as an important regulator of organogenesis [15]. The gene responsible for UMS was mapped to chromosome 12 at position 12q23-q24.1 in 1995 and was identified to be *TBX3* in 1997 [16,17]. It encompasses 7 exons and encodes a 723 amino acid protein [18]. The TBX3 protein consists of a DNA binding domain, also known as the T-box domain, a nuclear localization signal, an activation domain, and two repression domains (R1 and R2) [19]. Multiple isoforms of *TBX3* have been described with the TBX3 and TBX3 + 2a being the predominant isoforms [2]. The likely pathogenic variant segregating in this family is expected to produce an in-frame deletion within the T-box domain. Eighteen *TBX3* variants associated with UMS phenotype are described in the literature; four nonsense, eight frameshift, three missense, and three splice site variants [1,18,20] (Figure 2). The presence of significant intra- and inter-familial phenotypic heterogeneity suggests that there may be alternative mechanisms or genetic modifiers in addition to haploinsufficiency of the *TBX3* gene. Emerging evidence suggests that *TBX3* variants which affect the DNA binding and nuclear localization signal (NLS) region results in haploinsufficiency whereas variants in the C-terminus result in a dominant negative effect on splicing inhibition of the wildtype TBX3 [21]. The latter pathway results in alternative splicing, although the exact mechanism is unknown and will require extensive experiments to be performed.

The additional physical findings of congenital sagittal craniosynostosis in the proband’s father are likely secondary to the missense variant in the *EFNA4* (c.178C > T, p.His60Tyr). This variant was found in one patient with unilateral coronal suture synostosis, and segregation analysis showed an incompletely penetrant phenotype in the parents of the proband [22]. Subsequent studies identified the *EFNA4* variant (c.178C > T, p.His60Tyr) in three additional patients with nonsyndromic, single suture synostosis (Appendix A) [23,24]. Functional analysis by Merill et al. demonstrated that this variant results in a 65% loss of the binding of ephrin-A4 proteins to EphA7, the partner receptor, by disrupting the formation of the ligand–receptor heterotetramer [22]. This interruption of the Ephrin-Eph signaling prevents the normal restriction of the migration pathway at the interface of the frontal and parietal bones, resulting in premature closure of these sutures and the clinical finding of craniosynostosis. Although this variant is classified as likely benign in the ClinVar and Varsome databases as it is reported in healthy adults within the Genome Aggregation Database (gnomAD) exome allele frequency of 0.00135, this amino acid is highly conserved across species, has a high CADD score, and ten in silico protein prediction algorithms assess this variant to be damaging (DANN, DEOGEN2, EIGEN, FATHMM-MKL, LIST-S2, M-Cap, MVP, MutationAssessor, MutationTaster, and SIFT). A review of WGS data for 217 (92 coronal, 63 sagittal, and 62 metopic sutures) nonsyndromic craniosynostosis case-parent trios from the International Craniosynostosis Consortium did not reveal any variants that were pathogenic, likely pathogenic, or of unknown significance in *EFNA4* (unpublished data). According to the American College of Medical Genetics (ACMG) guidelines [25] for sequence variant interpretation, the pathogenic variant *EFNA4* (c.178C > T, p.His60Tyr) present in the father would fall in the category of a variant of unknown significance (evidence PS3, PM6, and PP3). However, we make the argument that it is leaning towards likely pathogenic rather than likely benign despite the higher-than-expected minor allele frequency. It is possible that the presence of this variant in healthy adults may be due to incomplete penetrance or variable expressivity of craniosynostosis with a milder phenotype or an incomplete sutural synostosis.

## 5. Conclusions

This family further extends the genotypic spectrum of UMS and highlights the possibility of concurrent variants in additional genes when unexpected phenotypes are present. The UMS phenotypic variability is rather marked in this family from asymptomatic to multisystemic presentation. Further studies investigating the functional consequence of this novel splicing mutation in *TBX3* will be useful in expanding our knowledge of developmental biology and interacting pathways in embryogenesis. The presence of the missense variant in *EFNA4* also adds to the phenotypic spectrum of nonsyndromic craniosynostosis and reclassification of this variant should be considered as it has a significant impact on genetic counseling for reproductive purposes for families.

## Figures and Tables

**Figure 1 genes-13-01649-f001:**
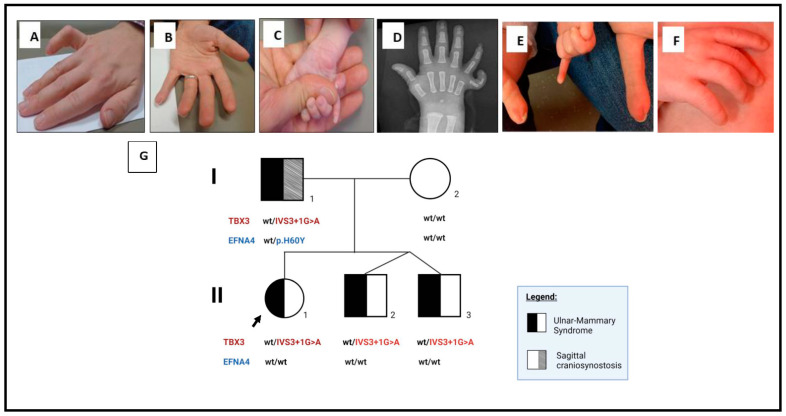
Phenotypic and genotypic findings in the proband and her family. (**A**,**B**) Clinical findings in the proband’s father showing a contracture of the fifth finger and duplication of the fifth fingernail. (**C**,**D**) The proband’s right hand showing the postaxial polydactyly and duplication of the fifth fingernail and the corresponding X-ray of the proband’s right hand. (**E**) Unilateral, fifth finger contracture in one of the proband’s twin brothers. (**F**) Normal hand in the other twin brother. (**G**) Pedigree of the family with the associated *TBX* and *EFNA4* allele. The proband is indicated by the short arrowhead in black (Individual II:1). The proband and her siblings (II:2 and II:3) are heterozygous for the *TBX3* variant. The proband’s father (Individual I:1) is heterozygous for both the *TBX3* and *EFNA4* variants. The proband’s mother (Individual 1:2) is wild-type for both alleles.

**Figure 2 genes-13-01649-f002:**
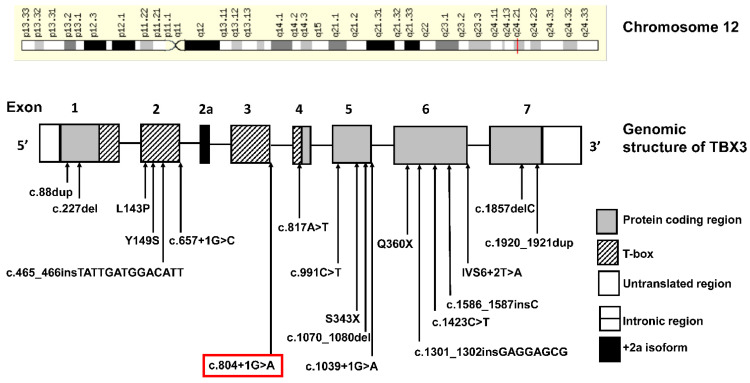
Chromosomal position and genomic structure of the T-Box 3 (*TBX3*) gene and known disease-causing variants which have been published in the literature. Our proband’s variant is highlighted in the red box.

## Data Availability

The data from this study are not publicly available due to privacy or ethical restrictions but are available on request from the corresponding author.

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
