# Peer review of "TBX3 and EFNA4 Variant in a Family with Ulnar-Mammary Syndrome and Sagittal Craniosynostosis"

_genes, 2022, doi:10.3390/genes13091649_

Round 1

Reviewer 1 Report

The article is well written and concise. The complex clinical presentation of the family is interesting and worth be published, helping on rare variant interpretation and improving clinical description of rare conditions. The splicing variant has not been functionally validated at the mRNA level.

Detailed Case Description:

Page 2, line 71-73: The effect of the splicing variant has to be further explained (in a materials and methods section?), including which algorithm has been used. A figure with the putative alternative donor sites and their scores could be of great utility to support this affirmation. Even then, this sentence must be tone down as this is not a functional validation at the mRNA level.

Page 2, line 82: please, indicate the total number of carriers. The maximum allele frequency for this variant is quite high (0.002 is “high” for an AD variant). In addition, 127 controls in gnomAD (v2) carry this variant, including 2 homozigotes. These frequencies are even higher in v3 of gnomAD. This could be discussed in more detail in the discussion section.

Figure 1: I suggest to move the chromatograms to supplementary data and to include the genotype of each allele in the pedigree for easier interpretation of the data.

Discussion section:

I do not agree that PM2 and PP1 criteria can be applied for the variant classification. The maximum MAF is quite high and the variant do not segregate in the family (only the father presents with craniosynostosis so there aren’t multiple individuals to test that). On the other hand, PS3 can be applied as the effect of the variant has been functionally tested. Overall, I do agree that the variant is a good candidate to explain the father’s craniosynostosis and could be a recurrent variant, quite common in general population.

The materials and methods section is missing and should be included with a description of the NGS methodologies and variant filtering and prioritization. It is evident that a strict clinical filtering has been applied on the probands father WGS, so, the list of genes analyzed should be included (as in Suppl. Table 1). I also encourage the authors to include a VUS list as suppl. Material. Some other variants may be modifying the phenotype and could improve future knowledge, helping to understand incomplete penetrance and variable expression. Also, the authors mentioned some predictions regarding the splice variant that are not described anywhere, this information must be included.  

Reviewer 2 Report

The authors presented a genetic study of a family with ulnar-mammary syndrome and sagittal craniosynostosis. Overall, this work is of interest to researchers working in the field of developmental anomalies. However, the work also raises several questions and there are areas that could benefit from further clarification.

I have some comments as below.

1.     P.1, lines 38-39. Reference 2. The article summarizes the data up to 1999. According ORPHANET, “Up to date, approximately 128 cases of Ulnar-mammary syndrome (UMS) have been reported in the literature” https://www.orpha.net/consor/cgi-bin/Disease_Search.php?lng=EN&data_id=2808&Disease_Disease_Search_diseaseGroup=Ulnar-mammary-syndrome&Disease_Disease_Search_diseaseType=Pat&Disease(s)/group%20of%20diseases=Ulnar-mammary-syndrome&title=Ulnar-mammary%20syndrome&search=Disease_Search_Simple.

2.     Figure 1B. Proband’s grandparents are included in the panel. Please add any information about grandparents. If there is no information, should they be indicated in the figure?

3.     Is it possible to include data on the ethnic origin of the proband in order to correlate these data with the GnomAD allele frequencies in the corresponding population?

4.     P.2, lines 71-73. Please provide explanation about in silico model prediction.

5.     P.5, line 163. Please use italics when writing the name of the gene.

6.      S. Figure 1. There is no metopic suture in S. Figure1.
